# Cervical cancer screening outcomes for HIV-positive women in the Lubombo and Manzini regions of Eswatini—Prevalence and predictors of a positive visual inspection with acetic acid (VIA) screen

Rufaro Mapaona[1], Victor Williams[1,2]*, Normusa Musarapasi[1], Sharon Kibwana[3], Thokozani Maseko[1], Rhinos Chekenyere[1], Sidumo Gumbo[1], Phetsile Mdluli[1], Hugben Byarugaba[1], Dileepa Galagedera[1], Arnold Mafukidze[1], Alejandra Hurtado-de-Mendoza[4], Prajakta Adsul[5], Pido Bongomin[1], Christopher Loffredo[4], Xolisile Dlamini[6], Deus Bazira[3], Sylvia Ojoo[3], Samson Haumba[1,3]

**1** Center for Global Health Practice and Impact, Georgetown University, Mbabane, Eswatini, **2** Julius Global Health, Julius Center for Health Sciences and Primary Care, University Medical Center Utrecht, Utrecht University, Utrecht, The Netherlands, **3** Center for Global Health Practice and Impact, Georgetown University Medical Center, Washington, DC, United States of America, **4** Lombardi Comprehensive Cancer Center, Georgetown University, Washington, DC, United States of America, **5** University of New Mexico Comprehensive Cancer Center, Albuquerque, New Mexico, United States of America, **6** National Cancer Control Program, Ministry of Health, Mbabane, Eswatini

* vmw20@georgetown.edu

**Data Availability Statement:** The data underlying this research is readily available and can be

## Abstract

This study aimed to describe the prevalence and predictors of a positive VIA (visual inspection with acetic acid) cervical cancer screening test in women living with human immunodeficiency virus (HIV). We retrospectively analysed data from women aged ≥15 who accessed VIA screening from health facilities in the Lubombo and Manzini regions of Eswatini. Socio-demographic and clinical data from October 2020 to June 2023 were extracted from the client management information system (CMIS). VIA screening outcome was categorised into negative, positive, or suspicious. A logistic regression model estimated the adjusted odds ratio (AOR) of the predictors of a positive VIA screen at p<0.05 with 95% confidence intervals. Of 23,657 participants, 60.8% (n = 14,397) were from the Manzini region. The mean age was 33.3 years (standard deviation 7.0), and 33% (n = 7,714) were first-time screens. The prevalence of a positive VIA was 2.6% (95% CI: 2.2%, 3.0%): 2.8% (95% CI: 2.2%, 3.5%) in Lubombo and 2.4% (95% CI: 2.0%, 2.9%) in Manzini (p = 0.096). Screening at mission-owned (AOR 1.40; p = 0.001), NGO-owned (AOR 3.08; p<0.001) and industrial/workplace-owned health facilities (AOR 2.37; p = 0.044) were associated with increased odds of a positive VIA compared to government-owned health facilities. Compared to those aged 25–34, the odds of a positive VIA increased by 1.26 for those in the 35–44 age group (AOR 1.26; p = 0.017). Predictors with lower odds for a positive VIA test were: being on anti-retroviral therapy (ART) for 5–9 years (AOR 0.76; p = 0.004) and ≥10 years (AOR 0.66; p = 0.002) compared to <5 years; and having an undetectable viral load (AOR 0.39; p<0.001)

accessed at https://doi.org/10.6084/m9.figshare.25288507.

**Funding:** The Support Eswatini Achieve and Sustain HIV Epidemic Control (SEASEC) Program is supported by the U.S President's Emergency Plan for AIDS Relief through the Centers for Disease Control and Prevention (Co-operative Agreement No.:NU2GGH002294), implemented by Georgetown University in collaboration with the Government of the Kingdom of Eswatini. The funders had no role in the conceptualization of the study, study design, data collection and analysis, decision to publish and preparation of the manuscript.

**Competing interests:** The authors have declared no competing interest exist.

compared to unsuppressed. Longer duration on ART and an undetectable viral load reduced the odds, while middle-aged women and screening at non-public health facilities increased the odds of a positive VIA screen.

## Introduction

Cervical cancer continues to be a leading cause of morbidity and mortality for women, despite being preventable by vaccination against the human papillomavirus (HPV) and curable if detected and treated early by screening [1]. There were an estimated 604,000 new cervical cancer cases and 342,000 deaths worldwide in 2020 [2], and it is the fourth most frequently diagnosed cancer in women [3].

There are substantial inequalities in the global cervical cancer burden: incidence is three times higher in countries with lower levels of human development [4], and more than 90% of deaths occur in these countries. The highest regional incidence and mortality occurs in Sub-Saharan Africa, where age-standardised incidence rates in Eastern Africa (40.1), Southern Africa (36.4) and Central Africa (31.6) [2] are far higher than the threshold of 4 per 100,000 established by the World Health Assembly's Global Strategy for cervical cancer elimination [3].

Most cervical cancer cases (99.7%) are caused by persistent HPV infection [5], which is more prevalent in HIV-infected women [6]. The natural history of HPV infection has a slow, 10-15-year progression to pre-cancer in immuno-competent people. In HIV-infected women, the condition progresses more frequently and quickly [7]. Cervical cancer is classified as an AIDS-defining illness. It is associated with lower CD4 cell counts and a lack of anti-retroviral therapy (ART) among women living with HIV [8]. Globally, approximately 1 in 20 cervical cancers is attributable to HIV. In sub-Saharan Africa, about 1 in 5 cervical cancers is due to HIV, which threatens the gains that improving access to HIV care and treatment has made in prolonging the life expectancy of these women. However, it is essential to note that although cervical pre-cancer among women living with HIV is common, those who receive regular cervical screenings among this population have low incidence rates of invasive cervical cancer [9].

Disparities in cervical cancer incidence and mortality reflect unequal access to and coverage of comprehensive prevention (including HPV vaccination), screening and treatment of precancerous lesions, and diagnosis and treatment of invasive cancers. The availability of these services is suboptimal in low- and middle-income countries, including Eswatini [10, 11]. As of 2020, less than 30% of lower-middle-income countries had introduced the HPV vaccine, compared to 85% in high-income countries [12, 13]. Still, in 2020, less than 35% of low-income countries had national cervical cancer screening programs, and less than 30% reported the availability of pathology services, cancer surgery, and other cancer management services [14].

Secondary prevention reduces cervical cancer incidence and mortality by identifying and treating women with pre-cancerous lesions. The World Health Organization (WHO) recommends HPV DNA testing as the primary screening test for cervical cancer, but this technology is not yet available or accessible in many countries [13]. Cytology-based (or pap smear) screening has been successful when implemented as part of national programmes with high coverage and in settings where resources exist for patient follow-up, additional diagnostic tests (colposcopy and pathology), and disease management [15]. In low-resource settings, the "see-and-treat" approach, which includes naked eye or digitally enhanced visual inspection with acetic acid (VIA) to detect pre-cancerous lesions and the use of cryotherapy (liquid nitrous oxide ablation) to freeze and destroy pre-cancerous tissue, has been successfully implemented [16].

However, the quality of VIA depends heavily on provider competence and the test's sensitivity, which is variable [17–19].

Prevalence of VIA positivity in HIV-positive women is variable across studies ranging between 2–17%, but this may be dependent on the competence and experience of the service provider rather than disease prevalence [20–27]. In a study conducted in Eswatini, the positivity ranged between 6.3 and 25.1% [28] and was dependent on training and refresher courses. VIA positivity decreased with improved competence of the service providers, suggesting that training and refreshers reduced false positive tests. In a study in Tanzania, positivity was highest after training [25]. In another study, the presence of cervical lesions was greater in WLHIV (22,9%) compared to their HIV-negative counterparts [27]. Studies from Nigeria and India showed similar variability amongst healthcare providers. VIA positivity ranged from 0–25% when conducted by nurses in Nigeria [29] and 4–31% when conducted by six different gynaecologists in India [25].

Predictors for VIA positivity were older age group, single marital status (widowed/divorced), age at first sexual intercourse, number of lifetime sexual partners, and low level of education across a range of prior studies [20, 23, 25, 28].

## Cervical cancer in Eswatini

Eswatini faces a high dual burden of HIV and cervical cancer [30]. The country's HIV prevalence of 24.8% among adults is one of the highest in the world [31], and Eswatini has the highest age-standardised cervical cancer incidence rate of 84.5 per 100,000. Despite a high ART coverage and viral load suppression of ≥95% amongst ART clients [32], only 17.6% of the country's estimated 336,037 women aged 15–64 years have been screened at least once for cervical cancer [33]. Coverage is only slightly better for women living with HIV, and estimates suggest that as of June 2023, only 20.3% of this population has been screened in the Lubombo and Manzini regions [34].

Given the high dual burden of HIV and cervical cancer in the country, the Eswatini Ministry of Health has developed several policies. In particular, the National Cancer Prevention and Control Strategy of 2019 has set clear targets to increase the percentage of health facilities providing screening, early detection, and linkage to treatment for all cancers to 60% by 2022. However, only 41% of health facilities in Eswatini have achieved this goal [35]. Current guidelines recommend screening as soon as one is sexually active for HIV-positive women. In contrast, HIV-negative women or those with unknown status can start screening from ≥25 years of age. The recommended frequency of cervical cancer screening is also based on HIV status and HPV infection status of patients (if known). Women living with HIV are screened annually, while it is every two years for HIV-negative women and extended to three years for those who are HIV and HPV-negative [36]. All screening services are free in public hospitals. HPV DNA testing is not yet available in the country. In this study, we describe the cervical cancer screening outcomes, prevalence and predictors of a positive cervical screen among HIV-positive women accessing services at health facilities in two regions of Eswatini. We also provide recommendations to strengthen cervical screening services in line with the various existing Ministry of Health (MOH) policies to meet the needs of women living with HIV.

## Methods

### Ethics statement

This study is covered under the protocol approved by the Eswatini Health and Human Research Review Board (EHHRRB 116/2022) for Georgetown University to analyse program data for dissemination. Additional approval has been obtained from the Georgetown

University Institutional Review Board (GU—IRB) (STUDY 00006034) and the United States Centers for Disease Control (CDC) (Accession #: CGH-ESW-9/14/23-15af6). Anonymised data were used to ensure confidentiality, and the EHHRRB also approved an application for a waiver of written informed consent from participants since the data is from routine care extracted from the electronic medical records.

### Study design and setting

This was a retrospective cohort study among HIV-positive women receiving HIV care at all health facilities providing HIV treatment services and accredited to provide cervical cancer screening services in the Lubombo and Manzini regions of Eswatini. These two regions are where the United States Government funded Support Eswatini Achieve Sustained HIV Epidemic Control (SEASEC) project is implemented. The SEASEC project is a comprehensive HIV care and treatment program implemented by the Eswatini Ministry of Health (MOH) in partnership with Georgetown University. SEASEC supports the provision of cervical cancer screening and treatment of pre-cancerous lesions for all women living with HIV. As part of the program, cervical cancer screening and treatment services have been established through a network of seventy-five health facilities in Lubombo and Manzini. Healthcare workers have been trained to offer VIA, cryotherapy, and Loop Electrosurgical Excision Procedure (LEEP) services. Screening supplies and equipment are also provided. Additional interventions include support for patient referral within the network, quality improvement, and collection and use of service delivery data.

Health facilities in the Lubombo and Manzini regions are described in Table 1. Lubombo region has fifty-four health care facilities, including two hospitals (with in-patient services), one rural health centre, one public health unit (equivalent to a multi-department health centre), and forty-two primary health clinics. Manzini region has one hundred and twenty-three health facilities, including four hospitals (with in-patient services), two public health units (equivalent to health centres), and one hundred and seventeen primary health clinics. The SEASEC Program supports eighty-six public, private or faith-based health facilities across both regions (Table 1).

Fig 1 shows the locations of the 75 health facilities that implement the SEASEC Program-supported cervical screening activities.

### Study participants, sample size and sampling

We generated data for the study from patient attendance records from October 2020 to June 2023. The data are collected using standardised service registers provided by the Eswatini Ministry of Health (MOH) to all public health facilities. Our study participants were HIV-positive women screened for cervical cancer from October 2020 to June 2023 at any of the 75 health facilities described above. Data from the two regions indicate that, as of June 2023, a total of 69,529 women aged 15 years or older were living with HIV, and approximately 19,616 of these had received at least one cervical cancer screening [34]. During data extraction, the data of 23,694 women who received a cervical screen were included in the analysis.

### Data sources and study variables

Data were extracted from the National Electronic Medical Record: the Client Management Information System (CMIS) on the 2nd of August 2023. The required variables were filtered and extracted per the study objectives. The variables describe the patient's sociodemographic and clinical information (Table 2). Age was grouped into eight categories using 5-year age bands from 15 to 50 years, while parity was categorised into 0, 1, 2–3, 4–5, and ≥6. Marital

**Table 1. Health facilities in Manzini and Lubombo regions (*Obtained from Manzini and Lubombo Regional health management team quarterly report, Q2 2023*).**

| Health Facility Type | Manzini Region | | | Lubombo Region | | | Description of health facilities |
|---|---|---|---|---|---|---|---|
| | *Total Health Facilities* | SEASEC Supported | | *Total Health Facilities* | SEASEC Supported | | |
| | | Total | Cervical services | | Total | Cervical services | |
| Government | 41 | 26 | 26 | 30 | 25 | 22 | • No fees for all services<br>• Receives PEPFAR* support for HIV/TB services<br>• Receives HIV/TB supplies from MOH |
| Faith-based | 19 | 13 | 7 | 9 | 9 | 9 | • Owned by a faith-based entity<br>• Low user fees<br>• Free HIV/TB services<br>• Receives government support<br>• Receives PEPFAR support for HIV/TB services<br>• Receives HIV/TB supplies from MOH |
| NGO | 11 | 5 | 5 | 2 | 2 | 1 | • Owned by an NGO<br>• No user fees<br>• Mainly HIV/TB services<br>• Receives PEPFAR support for HIV/TB services<br>• Receives HIV/TB supplies from MOH |
| Industrial/ Workplace | 0 | 0 | 0 | 7 | 7 | 5 | • Owned by the company<br>• No user fees for employees<br>• Non-employees pay user fees<br>• Receives PEPFAR support for HIV/TB services<br>• Receives HIV/TB supplies from MOH |
| Private | 52 | 0 | 0 | 6 | 0 | 0 | • Owned by private entities<br>• Services vary (general medical to specialised)<br>• Fees for all services |
| **Total** | **123** | **44** | **38** | **54** | **42** | **37** | |

*United States Presidents Emergency Plan for AIDS Relief (The US Government supports SEASEC through PEPFAR)

status was grouped into two classes: married/living with a partner or single. The patient's last viral load was categorised into three groups–undetectable ($<50$), suppressed ($50<1000$), and unsuppressed ($\geq 1000$). The timing of the cervical screening was divided into three groups–first-time screening, post-treatment follow-up screening, and rescreening per the national guidelines. The VIA screening outcome was categorised into negative, positive, and suspicious. The outcome variable for this study is the prevalence of a positive VIA screen, defined as the proportion of women with a positive VIA screen.

## Statistical analysis

Data was extracted in.xls format and imported into Stata 17 (College Station, TX) for cleaning and analysis. Descriptive analysis summarised underlying trends and seasonal patterns. Key patient characteristics are presented in Table 2, disaggregated by the cervical screening outcome. The outcome variable had no missing observations. Cases with missing observations for the predictor variables were excluded during analysis. The prevalence of positive VIA screens

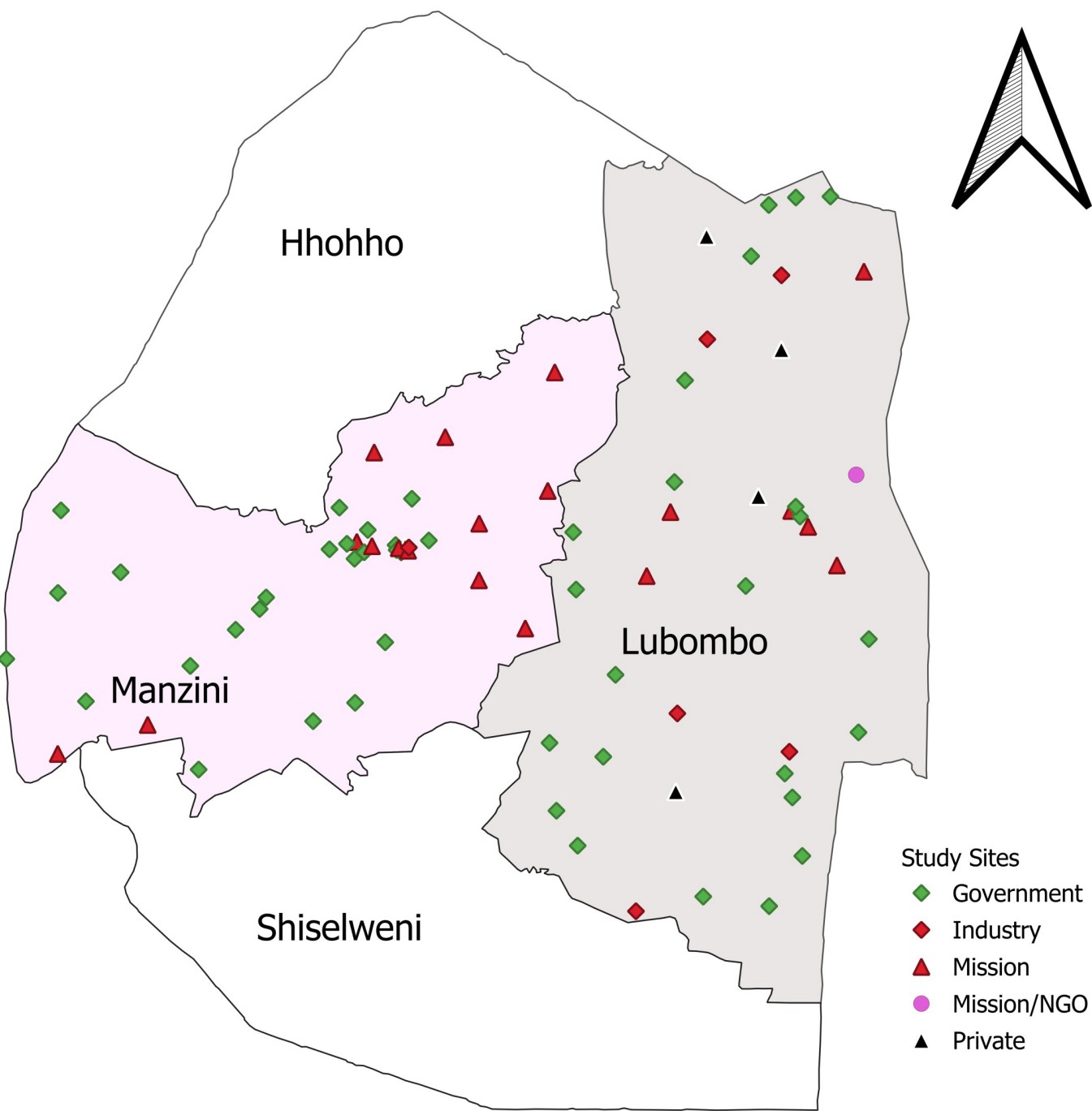

**Fig 1. Distribution of health facilities providing cervical cancer services in the Lubombo and Manzini region supported by the SEASEC program (Generated using SEASEC program data) (https://data.humdata.org/dataset/cod-ab-swz; https://www.esri.com/about/newsroom/announcements/esri-provides-open-access-to-key-federal-geospatial-data/).**

was presented overall and disaggregated by other patients' sociodemographic and clinical information.

A new binary dependent variable–presence or absence of cervical lesion—was generated to determine the predictors of a positive cervical screen. A logistic regression model was

**Table 2. Descriptive characteristics of participants.**

| Patient Characteristics | Negative (N = 22957) | Positive (N = 603) | Suspicious (N = 97) | Total (N = 23657) | P-Value* |
|---|---|---|---|---|---|
| **Region** | | | | | |
| Lubombo | 8976 (39.1%) | 256 (42.5%) | 39 (40.2%) | 9297 (39.2%) | 0.244 |
| Manzini | 13981 (60.9%) | 347 (57.5%) | 58 (59.8%) | 14397 (60.8%) | |
| **Age (years)** | | | | | |
| 15–19 | 284 (1.2%) | 11 (1.8%) | 0 (0.0%) | 295 (1.2%) | |
| 20–24 | 2123 (9.2%) | 54 (9.0%) | 2 (2.1%) | 2194 (9.3%) | |
| 25–29 | 4972 (21.7%) | 108 (17.9%) | 14 (14.4%) | 5098 (21.5%) | |
| 30–34 | 5776 (25.2%) | 147 (24.4%) | 24 (24.7%) | 5949 (25.1%) | 0.001 |
| 35–39 | 5268 (22.9%) | 155 (25.7%) | 24 (24.7%) | 5448 (23.0%) | |
| 40–44 | 3138 (13.7%) | 97 (16.1%) | 18 (18.6%) | 3268 (13.8%) | |
| 45–49 | 1215 (5.3%) | 27 (4.5%) | 14 (14.4%) | 1256 (5.3%) | |
| ≥50 | 181 (0.8%) | 4 (0.7%) | 1 (1.0%) | 186 (0.8%) | |
| Mean (SD) | 33.29 (7.06) | 33.53 (7.06) | 36.47 (6.78) | 33.31 (7.06) | <0.001 |
| **Parity** | | | | | |
| 0 | 9078 (55.9%) | 238 (57.2%) | 33 (51.6%) | 9373 (55.9%) | |
| 1 | 2295 (14.1%) | 61 (14.7%) | 14 (21.9%) | 2377 (14.2%) | |
| 2–3 | 3687 (22.7%) | 89 (21.4%) | 12 (18.8%) | 3789 (22.6%) | 0.005 |
| 4–5 | 1010 (6.2%) | 22 (5.3%) | 1 (1.6%) | 1038 (6.2%) | |
| ≥6 | 179 (1.1%) | 6 (1.4%) | 4 (6.3%) | 189 (1.1%) | |
| Median (Q1, Q3) | 0.0 (0.0, 2.0) | 0.0 (0.0, 2.0) | 0.0 (0.0, 2.0) | 0.0 (0.0, 2.0) | 0.626 |
| **Duration on ART (Years)** | | | | | |
| Median (Q1, Q3) | 6.0 (4.0, 8.0) | 6.0 (3.0, 9.0) | 7.0 (4.0, 8.0) | 6.0 (4.0, 9.0) | 0.390 |
| **Facility Type** | | | | | |
| Government | 12666 (55.2%) | 260 (43.1%) | 49 (50.5%) | 13003 (54.9%) | |
| Faith-based | 6760 (29.4%) | 180 (29.9%) | 32 (33.0%) | 6981 (29.5%) | <0.001 |
| NGO | 3420 (14.9%) | 157 (26.0%) | 11 (11.3%) | 3588 (15.1%) | |
| Industrial/Workplace | 111 (0.5%) | 6 (1.0%) | 5 (5.2%) | 122 (0.5%) | |
| **Cervical Screening Type** | | | | | |
| First Time Screening | 7427 (32.4%) | 244 (42.0%) | 29 (30.9%) | 7714 (32.6%) | |
| No Outcome | 10 (0.0%) | 22 (3.8%) | 2 (2.1%) | 34 (0.1%) | <0.001 |
| Post Tx FU | 70 (0.3%) | 34 (5.9%) | 1 (1.1%) | 105 (0.4%) | |
| Rescreening | 15432 (67.3%) | 281 (48.4%) | 62 (66.0%) | 15798 (66.8%) | |
| **Marital status** | | | | | |
| Married/living with partner | 4455 (30.5%) | 131 (34.8%) | 30 (56.6%) | 4634 (30.7%) | <0.001 |
| Single | 10164 (69.5%) | 245 (65.2%) | 23 (43.4%) | 10451 (69.3%) | |
| **Last VL Outcome (cells/ml)** | | | | | |
| Undetectable (<50) | 19505 (85.0%) | 469 (77.8%) | 84 (86.6%) | 20095 (84.8%) | |
| Suppressed (50<1000) | 2389 (10.4%) | 97 (16.1%) | 8 (8.2%) | 2494 (10.5%) | <0.001 |
| Unsuppressed (≥1000) | 406 (1.8%) | 25 (4.1%) | 3 (3.1%) | 434 (1.8%) | |
| No VL Result | 657 (2.9%) | 12 (2.0%) | 2 (2.1%) | 671 (2.8%) | |

fitted using the dependent variable and sociodemographic and clinical variables as predictors. A univariable model was initially built using each of the independent variables, and a stepwise forward and backward elimination process was used to select variables for inclusion in the final multivariable model at p = 0.20. The final predictors were determined from the multivariable model at p<0.05. Given the large sample size, calibration and discrimination

of the final logistic model were assessed using the *pmcalplot* command in Stata 17, which reports calibration-in-the-large, calibration slope, and area under the curve (AUC) [37]. Calibration-in-the-large is the difference between the mean predicted and the mean observed probabilities, which should be zero, while the calibration slope assesses model fitness and should equal 1 [37, 38]. The model is underfitted if slope>1 and overfitted if slope<1. The AUC is identical to the c-statistic and describes the ability of a model to distinguish a patient with the outcome from those without it and should be >0.5 [38]. Multicollinearity between variables in the model was assessed using the variance inflation factor (VIF). Strengthening the Reporting of Observational Studies in Epidemiology (STROBE) statement guided the reporting of this study (S1 Table).

## Results

### Sociodemographic and screening outcome characteristics of study participants

Table 2 presents the descriptive characteristics of all HIV-positive patients included in the study and disaggregated by screening outcomes (negative, positive, and suspicious cervical screens). Of 23,694 patients' data extracted from the database, 37 (0.15%) without outcome measures were excluded, leaving 23,657 included in the analysis. Six hundred and three (2.6%) had a positive screen, and 60.8% (n = 14, 397) were from the Manzini region. The mean age was 33.3 years (SD 7.0). Analysis by 5-year categories showed that screening increased gradually from 1.2% (n = 295) in the 15–19 age group to 25% (n = 5949) in the 30–34 age group and was consistent for both positive and negative participants. Most patients (67.3%, n = 15,798) were rescreened, while 32.6% (n = 7,714) were screened for the first time. About 69% (n = 10, 451) were single, and more than half of the patients (55.9%, n = 9,373) had a parity of 0. This was similar across the three outcome groups. The majority (96.4%, n = 22,770) of the patients were on a dolutegravir-based ART regimen; the median duration of ART was 6.0 years (IQR 4.0, 9.0), and 95% overall had a suppressed and undetectable viral load. This is similar for those who screened positive (94%, n = 566).

### Prevalence of positive VIA screen

The overall prevalence of a positive cervical screen was 2.6% (95% CI: 2.2%, 3.0%): 2.8% (95% CI: 2.2%, 3.5%) in Lubombo and 2.4% (95% CI: 2.0%, 2.9%) in Manzini (p = 0.096). The prevalence was further analysed by age group, ART duration, parity, and cervical screening type (Table 3). The prevalence ranged from 2.1% (95% CI: 1.5%, 3.0%) in the 25–29 years age group to 3.7% (95% CI: 1.3%, 9.9%) in the 15–19 years age group, and it was statistically higher in the 40–44 years age group in Lubombo (4.2%; 95% CI: 2.5%, 6.8%) compared to those in Manzini (2.1%; 95% CI: 1.4%, 3.1%) (p = 0.001). Patients with parity ≥6 had an overall higher prevalence of 3.2% (95% CI: 0.5%, 19.5%), while those with parity = 0 had a higher prevalence in Lubombo compared to Manzini (p = 0.004). Prevalence was higher in patients receiving post-treatment follow-up screening at 32.7% (95% CI: 20%, 49%) and lowest in rescreening patients.

### Follow-up VIA screening

Two hundred and twenty-three of 603 patients with a positive first VIA result had follow-up VIA results. Of the 223, 125 (56%) had a negative result, 89 (40%) had a positive result, and 9 (4%) had a suspicious result. Only 21 of 97 patients with a suspicious result had a follow-up VIA result: 5 (23.8%) were negative, 3 (14.3%) were positive, and 13 (61.9%) remained

**Table 3. Prevalence of a positive cervical screen.**

| Patient Characteristics | n/N | % Prevalence (95% Confidence Interval) | | | P-value* |
|---|---|---|---|---|---|
| | | Lubombo | Manzini | Overall | |
| Region | - | 2.8% [2.2,3.5] | 2.4% [2.0,2.9] | 2.6% [2.2,3.0] | |
| Facility Type | | | | | |
| Government | 260/12926 | 2.1% [1.6,2.8] | 1.9% [1.4,2.6] | 2.0% [1.6,2.5] | **0.440** |
| Mission | 180/6760 | 4.7% [3.2,7.0] | 1.6% [1.0,2.6] | 2.6% [1.9,3.5] | **<0.001** |
| NGO | 157/3420 | 0.0% | 4.4% [3.5,5.6] | 4.4% [3.5,5.6] | - |
| Industrial/Workplace | 6/11 | 5.1% [2.3,10.9] | 0.0% | 5.1% [2.3,10.9] | - |
| Age group (years) | | | | | |
| 15–19 | 11/295 | 5.9% [1.4,21.7] | 2.3% [0.7,7.4] | 3.7% [1.3,9.9] | 0.108 |
| 20–24 | 54/2177 | 2.4% [1.1,5.2] | 2.6% [1.5,4.4] | 2.5% [1.6,3.9] | 0.789 |
| 25–29 | 108/5080 | 2.4% [1.3,4.2] | 2.0% [1.2,3.2] | 2.1% [1.5,3.0] | 0.389 |
| 30–34 | 147/5923 | 2.6% [1.5,4.4] | 2.4% [1.6,3.6] | 2.5% [1.8,3.4] | 0.632 |
| 35–39 | 155/5423 | 2.4% [1.4,4.1] | 3.1% [2.3,4.3] | 2.9% [2.2,3.8] | 0.136 |
| 40–44 | 97/3235 | 4.2% [2.5,6.8] | 2.1% [1.4,3.1] | 3.0% [2.1,4.2] | **0.001** |
| 45–49 | 27/1242 | 2.1% [1.0,4.7] | 2.2% [1.3,3.6] | 2.2% [1.4,3.4] | 0.932 |
| $\geq$50 | 4/185 | 4.9% [1.8,12.3] | 0.0% | 2.2% [0.8,5.6] | - |
| Duration on ART | | | | | |
| <5 years | 228/7655 | 3.2% [2.2,4.8] | 2.8% [2.2,3.7] | 3% [2.4,3.7] | 0.332 |
| 5-9years | 270/11418 | 2.5% [1.7,3.6] | 2.3% [1.7,3.1] | 2.4% [1.9,3.0] | 0.425 |
| > = 10years | 105/4487 | 2.8% [1.8,4.4] | 2.0% [1.4,2.8] | 2.3% [1.7,3.2] | 0.070 |
| Parity | | | | | |
| 0 | 238/9316 | 3.2% [2.2,4.6] | 2.2% [1.6,3.0] | 2.6% [2.0,3.2] | **0.004** |
| 1 | 61/2356 | 3.1% [1.6,5.9] | 2.2% [1.1,4.4] | 2.6% [1.6,4.1] | 0.184 |
| 2–3 | 89/3776 | 2.7% [1.4,5.0] | 2.1% [1.2,3.6] | 2.4% [1.6,3.5] | 0.272 |
| 4–5 | 22/1032 | 2.0% [0.8,5.0] | 2.3% [0.7,7.0] | 2.1% [1.0,4.4] | 0.768 |
| $\geq$6 | 6/185 | 4.7% [0.7,26.4] | 0.0% | 3.2% [0.5,19.5] | 0.097 |
| Frequency of screening | | | | | |
| First Time Screening | 244/7671 | 4.0% [2.4,6.5] | 2.9% [2.2,3.9] | 3.2% [2.5,4.1] | 0.068 |
| Post Treatment Follow-Up | 34/104 | 37.1% [16.5,64] | 30.4% [15.6,50.9 | 32.7% [20,49.0] | 0.491 |
| Rescreening | 281/15713 | 1.9% [1.4,2.6] | 1.7% [1.3,2.1] | 1.8% [1.5,2.2] | 0.235 |

*Compares Lubombo and Manzini

suspicious. Of 22,957 patients with a negative VIA result, follow-up screening data was available for 8403 patients: 8,288 (98.6%) remained negative, 50 (0.6%) were positive, and 13 (0.15%) had a suspicious result.

## Predictors of a positive cervical screen

The predictors of a positive cervical screen are presented in Table 4. In the univariable analysis, women screening at mission-owned, NGO-owned, and industrial/workplace-owned health facilities had higher odds of a positive cervical cancer screening result than those screening at a government-owned health facility. Being on ART for 5–9 years and $\geq$10 years compared to being on ART for less than five years and having an undetectable viral load compared to an unsuppressed viral load reduced the odds of a positive cervical screen.

In the multivariable analysis, residing in the Manzini region (AOR 0.55; 95% CI: 0.45, 0.67; p<0.001) compared to Lubombo; being on ART for 5–9 years duration (AOR 0.76; 95% CI:

**Table 4. Univariable and multivariable predictors of abnormal cervical screen.**

| Predictor | n/N | Univariable | | Multivariable | |
|---|---|---|---|---|---|
| | | OR (95% CI) | P-Value | AOR (95% CI) | P-Value |
| Region | | | | | |
| Lubombo | 256/9232 | 1 | | 1 | |
| Manzini | 347/14328 | 0.87 (0.74, 1.02) | 0.096 | 0.55 (0.45, 0.67) | **<0.001** |
| Health Facility Type | | | | | |
| Government | 260/12926 | 1 | | 1 | |
| Mission | 180/6760 | 1.30 (1.07, 1.57) | **0.008** | 1.40 (1.15, 1.72) | **0.001** |
| NGO | 157/3420 | 2.24 (1.83, 2.74) | **<0.001** | 3.08 (2.41, 3.95) | **<0.001** |
| Industrial/Workplace | 6/11 | 2.63 (1.15, 6.04) | **0.022** | 2.37 (1.02, 5.49) | **0.044** |
| Age (years) | | | | | |
| 25–34 | 255/11003 | 1 | | 1 | |
| 15–24 | 65/2472 | 0.88 (0.67, 1.16) | 0.358 | 1.04 (0.78, 1.39) | 0.777 |
| 35–44 | 252/8658 | 1.11 (0.84, 1.46) | 0.459 | 1.26 (1.04, 1.53) | **0.017** |
| 45+ | 31/1427 | 0.82 (0.53, 1.27) | 0.376 | 0.83 (0.55, 1.24) | 0.361 |
| Duration on ART (Years) | | | | | |
| <5 years | 228/7655 | 1 | | 1 | |
| 5–9 years | 270/11418 | 0.79 (0.66, 0.94) | **0.009** | 0.75 (0.62, 0.92) | **0.004** |
| = > 10 years | 105/4487 | 0.78 (0.62, 0.99) | **0.038** | 0.66 (0.50, 0.85) | **0.002** |
| Marital status | | | | | |
| Married/Partner | 131/4586 | 1 | | | |
| Single | 245/10409 | 0.82 (0.66, 1.02) | 0.070 | | |
| Parity | | | | | |
| 0 | 238/9316 | 1 | | | |
| 1 | 61/2356 | 1.01 (0.76, 1.35) | 0.925 | | |
| 2–3 | 89/3776 | 0.92 (0.72, 1.18) | 0.511 | | |
| 4–5 | 22/1032 | 0.83 (0.53, 1.29) | 0.411 | | |
| ≥6 | 6/185 | 1.28 (0.56, 2.91) | 0.559 | | |
| Last Viral Load Outcome | | | | | |
| Unsuppressed (≥1000) | 25/431 | 1 | | 1 | |
| Undetectable (<50) | 469/19974 | 0.39 (0.26, 0.59) | **<0.001** | 0.39 (0.26, 0.59) | **<0.001** |
| Suppressed (50<1000) | 97/2486 | 0.66 (0.42, 1.04) | **0.071** | 0.66 (0.42, 1.04) | **0.071** |

0.62, 0.92; p = 0.004) and ≥10 years (AOR 0.66; 95% CI: 0.50, 0.85; p = 0.002) compared to less than five years; and having an undetectable viral load result (AOR 0.39; 95% CI: 0.26, 0.59; p<0.001) compared to an unsuppressed viral load had lower odds of a positive cervical screen. Compared to being screened at government-owned health facilities, the odds of a positive cervical screen were 1.4 times higher in those who screened at mission-owned health facilities (AOR 1.40; 95% CI: 1.15, 1.72; p = 0.001), thrice higher at NGO-owned health facilities (AOR 3.08; 95% CI: 2.41, 3.95; p<0.001), and 2.3 times higher at industrial/workplace-owned health facilities (AOR 2.37; 95% CI: 1.02, 5.49; p = 0.044). Similarly, the odds of a positive cervical screen were 1.26 times higher for women aged 35–44 (AOR 1.26; 95% CI: 1.04, 1.53; p = 0.017) than those aged 25–34. The output of the calibration plot of observed versus expected probabilities is presented in Fig 2 and reports a reasonable calibration of the model parameters with model discrimination for a positive VIA of 0.6. Assessment for multicollinearity yielded a mean-variance inflation factor of 1.17, suggesting very low multicollinearity between the variables in the final model.

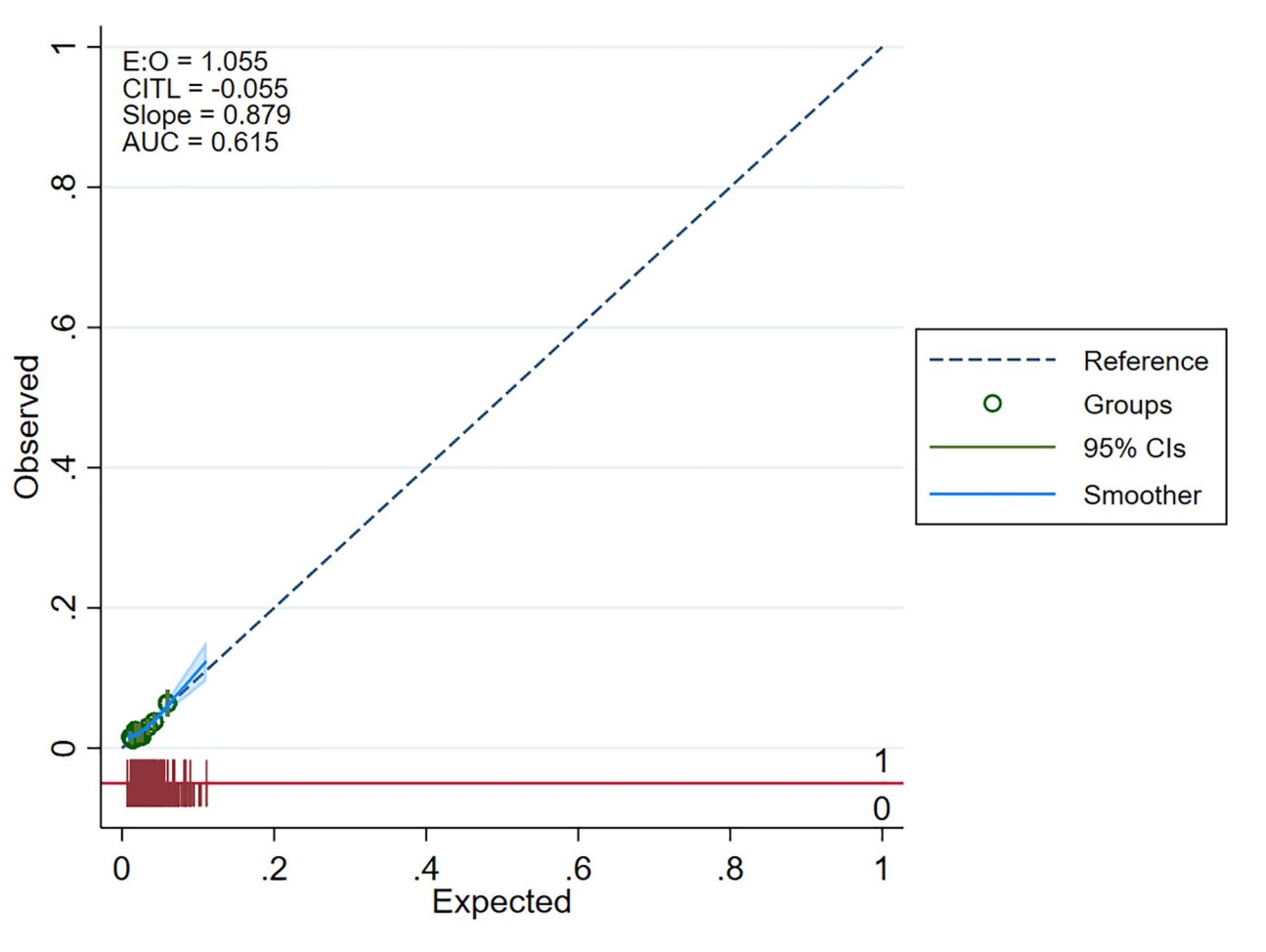

**Fig 2. Model calibration plot of observed versus expected probabilities.**

## Discussion

We aimed to describe the cervical cancer screening outcomes, prevalence, and predictors of a positive cervical screen in women living with HIV on ART who accessed services at health facilities providing cervical cancer screening in the Lubombo and Manzini regions of Eswatini. The prevalence of a positive cervical screen was 2.6%, similar across the two regions and highest in the 40–44 years age group in the Lubombo region. Those with a longer duration of ART and those with undetectable viral load had lower odds of a positive cervical screen. In contrast, women aged 35–44 had higher odds of a positive cervical screen. Our study findings suggest that the percent positivity for cervical cancer screening tests among WLHIV is lower than reported in the literature from studies in similar settings, regardless of the type of screening test used [21–23, 39, 40]. A previous cross-sectional study in Eswatini found that the presence of cervical lesions was higher in HIV-positive (22.9%) than HIV-negative women (5.7%; p < 0.0001) [28], and another study in Eswatini found a VIA positivity rate of 9% in the general population [28]. Elsewhere in Africa, a large cohort study of women in Zambia who were screened with VIA and digital cervicography (VIAC) had a positivity rate of 10.4% for the general population,

with WLHIV having a much higher rate of 53.3% [41]. In Nigeria, a study using program data found a VIA positivity rate of 7.1% [42]. In a hospital-based study conducted in Ethiopia, the prevalence of a pre-cancerous cervical lesion was 9.3% among WLHIV [43].

This finding of a lower-than-expected positivity rate in Eswatini compared to the studies cited above points to the need for further research into the screening offered in health facilities. We observed that the learning curve to implement this screening test is high, and the results during the initial phases often reflect lower positivity rates. Future research can help determine whether the lower-than-expected positivity rate is due to a need for strengthening screening services or is an accurate reflection of rates among this population. Given the move towards the recent WHO guidelines in Eswatini, HPV-DNA is being recommended for use as the primary screening test. Eswatini has yet to roll out this testing modality fully; hence, ensuring the quality of the existing VIA screening test is essential even as the government works to expand access to HPV DNA testing.

Another observation from our study is that the prevalence of a positive screening result was associated with accessing services at faith-based, NGO-owned and industrial/workplace-owned health facilities. This observation could be due to several reasons. First, personnel at these types of health facilities who provide cervical screening services are assigned to VIA screening as part of their ongoing, long-term responsibilities, which enables them to acquire skills and expertise over time. In comparison, health personnel at government-owned health facilities are not stationed at the VIA unit; they rotate from their role to another unit or health facility every twelve months. This rotation limits the skill and expertise in cervical screening. Secondly, the facilities with more positive screens have access to better screening equipment with better utilisation than government-owned facilities. Third, these health facilities enforce higher quality standards on VIA screening with strict adherence to the screening standard operating procedure (SOP) and supervision than government-owned health facilities. Fourth, they provide supplemental training for their staff and the required minimum training for health personnel providing VIA and cervical cancer services. All these collectively contribute to improved provider competence in delivering these screenings, as seen in previous literature [23, 25, 26, 39].

A positive screening result in our study was also associated with being aged between 35–44 years, compared to older and younger age groups. This finding is consistent with findings from several sub-Saharan African countries that the burden of cervical cancer attributable to HIV is highest among younger women aged less than 45 [44]. Longer duration on ART and an undetectable viral load were protective in our analysis, which is consistent with other studies and could be attributed to the increased immune function, which reduces the incidence and progression of squamous intraepithelial lesion (SIL) and cervical intraepithelial neoplasia (CIN) and, ultimately, the incidence of invasive cervical cancer when patients are stable on ART [45]. Other predictors of positive screening results vary in the literature. Studies have found that predictors of pre-cancerous cervical lesions or positive cervical cancer screening results among WLHIV include increasing parity [46], a history of multiple sexual partners, and sexually transmitted infections [47].

Our study had several strengths, notably the large sample size of nearly 24,000 women. Regionwide coverage of public, faith-based, NGO-owned and industrial/workplace-owned health facilities limits any possible effects of clinic-based selection bias. There was a high rate of data completion on the results of screening tests, with very few participants lost to follow-up or with missing data. Given our study's large number of health facilities, the findings highly represent the two regions. On the contrary, the study conclusions were limited in generalizability, as the SEASEC project supported all sites included in this study. These findings may not apply to settings not supported by the SEASEC project. This is also important for planning

future research to outline the role of the support partners, the characteristics of the settings, and the support received. Another limitation of our study is excluding private health facilities, which are not SEASEC-supported. A challenge with private health facilities includes difficulty accessing patient data, uncertainty about the type of services they offer, and whether they adhere to standard MOH guidelines. Our study did not present findings on clinical endpoints as the data was not readily accessible. Future studies on this cohort will include analysis of the clinical endpoints to identify any association with different patient characteristics. The final logistic model had low discrimination, but this was due to the need to achieve a more parsimonious model with a large sample size.

Eswatini has yet to achieve its target of reaching all WLHIV for cervical cancer services. Developing a better understanding of ongoing programmatic efforts can help promote the effective implementation of cervical cancer services to more health facilities with targeted, demand-driven activities and patient education, thereby increasing the cervical screening coverage in Eswatini. Study findings support the delivery of cervical cancer screenings in government-owned health facilities in Eswatini with measures to track adherence to cervical screening standards, standard operating procedures, supportive supervision, and mentoring for staff who require additional skills. Eswatini has recently included a focus on HPV vaccinations for eligible individuals [47], which has implications for future programmatic efforts in terms of cervical cancer screening. As implementation becomes country-wide, additional information may be needed to ascertain how the individual's vaccination status could impact VIA positivity. This information may have implications for the implementation of VIA guidelines as they do not consider the individual's vaccination status.

Finally, the study findings highlight the critical need for a focus on the implementation of cervical cancer screening services across the spectrum of clinics providing HIV and non-HIV-specific care. If supported, such studies could help provide implementation strategies and guidelines that can ensure the standardisation of services, focusing on improving healthcare capacity and health system structures required for a comprehensive screening program. In addition to WLHIV, such efforts can help prioritise screening for all women to reduce health and healthcare disparities.

## Supporting information

**S1 Table. STROBE checklist.**
(DOCX)

## Author Contributions

**Conceptualization:** Rufaro Mapaona, Victor Williams, Normusa Musarapasi, Sharon Kibwana, Rhinos Chekenyere, Sidumo Gumbo, Phetsile Mdluli, Hugben Byarugaba, Arnold Mafukidze, Pido Bongomin, Xolisile Dlamini, Sylvia Ojoo, Samson Haumba.

**Data curation:** Victor Williams, Thokozani Maseko, Hugben Byarugaba, Dileepa Galagedera, Arnold Mafukidze.

**Formal analysis:** Victor Williams, Thokozani Maseko, Dileepa Galagedera, Arnold Mafukidze.

**Funding acquisition:** Sharon Kibwana, Deus Bazira, Sylvia Ojoo, Samson Haumba.

**Investigation:** Rufaro Mapaona, Sidumo Gumbo.

**Methodology:** Rufaro Mapaona, Victor Williams, Normusa Musarapasi, Sharon Kibwana, Rhinos Chekenyere, Sidumo Gumbo, Hugben Byarugaba, Dileepa Galagedera, Arnold

Mafukidze, Alejandra Hurtado-de-Mendoza, Prajakta Adsul, Christopher Loffredo, Xolisile Dlamini, Deus Bazira, Sylvia Ojoo, Samson Haumba.

**Project administration:** Victor Williams, Thokozani Maseko, Phetsile Mdluli, Alejandra Hurtado-de-Mendoza, Prajakta Adsul, Christopher Loffredo.

**Resources:** Deus Bazira.

**Supervision:** Pido Bongomin, Xolisile Dlamini, Deus Bazira, Sylvia Ojoo, Samson Haumba.

**Validation:** Victor Williams, Normusa Musarapasi, Sharon Kibwana, Thokozani Maseko, Rhinos Chekenyere, Phetsile Mdluli, Hugben Byarugaba, Dileepa Galagedera, Arnold Mafukidze, Alejandra Hurtado-de-Mendoza, Prajakta Adsul, Pido Bongomin, Christopher Loffredo, Xolisile Dlamini, Deus Bazira, Sylvia Ojoo, Samson Haumba.

**Visualization:** Thokozani Maseko, Dileepa Galagedera.

**Writing – original draft:** Rufaro Mapaona, Victor Williams, Normusa Musarapasi, Sharon Kibwana, Thokozani Maseko, Rhinos Chekenyere, Sidumo Gumbo, Phetsile Mdluli, Hugben Byarugaba, Arnold Mafukidze, Alejandra Hurtado-de-Mendoza, Prajakta Adsul, Pido Bongomin, Christopher Loffredo, Sylvia Ojoo, Samson Haumba.

**Writing – review & editing:** Rufaro Mapaona, Victor Williams, Normusa Musarapasi, Sharon Kibwana, Rhinos Chekenyere, Phetsile Mdluli, Alejandra Hurtado-de-Mendoza, Prajakta Adsul, Pido Bongomin, Christopher Loffredo, Xolisile Dlamini, Deus Bazira, Sylvia Ojoo, Samson Haumba.

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
