## [Decision Letter · Decision Letter 0]

18 Jan 2024

PGPH-D-23-02447

Cervical Cancer Screening Outcomes for HIV-positive Women in the Lubombo and Manzini regions of Eswatini – Prevalence and Predictors of a Positive Visual Inspection with Acetic Acid (VIA) Screen

Dear Dr. Williams,

Thank you for submitting your manuscript to PLOS Global Public Health. After careful consideration, we feel that it has merit but does not fully meet PLOS Global Public Health’s publication criteria as it currently stands. Therefore, we invite you to submit a revised version of the manuscript that addresses the points raised during the review process.

We look forward to receiving your revised manuscript.

Kind regards,

Nebiyu Dereje, MPH, PhD

Academic Editor

Journal Requirements:

2. We ask that a manuscript source file is provided at Revision. Please upload your manuscript file as a .doc, .docx, .rtf or .tex.

3. Please provide separate figure files in .tif or .eps format only and remove any figures embedded in your manuscript file. Please also ensure all files are under our size limit of 10MB.

4. Some material included in your submission may be copyrighted. According to PLOS’s copyright policy, authors who use figures or other material (e.g., graphics, clipart, maps) from another author or copyright holder must demonstrate or obtain permission to publish this material under the Creative Commons Attribution 4.0 International (CC BY 4.0) License used by PLOS journals. Please closely review the details of PLOS’s copyright requirements here: PLOS Licenses and Copyright. If you need to request permissions from a copyright holder, you may use PLOS's Copyright Content Permission form.

Potential Copyright Issues:

Fig 1: please (a) provide a direct link to the base layer of the map (i.e., the country or region border shape) and ensure this is also included in the figure legend; and (b) provide a link to the terms of use / license information for the base layer image or shapefile. We cannot publish proprietary or copyrighted maps (e.g. Google Maps, Mapquest) and the terms of use for your map base layer must be compatible with our CC-BY 4.0 license. 

"

Additional Editor Comments (if provided):

- In the statistical analysis, please change multivariate analysis to multivariable analysis. Also, include how the model fitness was checked and multicollinearity handled.

- Please revise the interpretation of the predictors (multivariable analysis) in a way to indicate the strength of the association (not only the direction of the association).

- The Odds ratio in the multivariable analysis is the adjusted odds ratio (aOR), please indicate this in the table and text.

Reviewers' comments:

Reviewer's Responses to Questions

**Comments to the Author**

1. Does this manuscript meet PLOS Global Public Health’s publication criteria? Is the manuscript technically sound, and do the data support the conclusions? The manuscript must describe methodologically and ethically rigorous research with conclusions that are appropriately drawn based on the data presented.

Reviewer #1: Yes

Reviewer #2: Yes

2. Has the statistical analysis been performed appropriately and rigorously?

Reviewer #1: Yes

Reviewer #2: Yes

3. Have the authors made all data underlying the findings in their manuscript fully available (please refer to the Data Availability Statement at the start of the manuscript PDF file)?

Reviewer #1: Yes

Reviewer #2: Yes

4. Is the manuscript presented in an intelligible fashion and written in standard English?

Reviewer #1: Yes

Reviewer #2: Yes

5. Review Comments to the Author

Reviewer #1: Overall, the authors present a very nice paper characterizing VIA-based screening among women living with HIV in Eswatini. A few comments follow.

- As the authors note, the positivity rate is much lower in this study than in comparable populations and regions. While VIA has variable performance in most settings, it is still notable that the rate of positivity is 10x lower than a previous study in Eswatini, ~5x lower than in Zambia, and 3x lower than in Nigeria – let alone the rates of 40%+ in Burkina Faso and South Africa (all included in the discussion). I recommend further exploring any available data to build on the explanation of the low positivity rate, including trends over time to support the explanation that newly-introduced VIA yields lower positivity rates.

- The authors further include that one possible explanation for differences between sites included in the study is that some personnel were better trained to perform VIA. The differences in shift patterns between medical personnel in this system and in other systems is a reasonable contributor. However, to keep this discussion point in the paper, I would recommend that the authors include clinical endpoints following screening. Better trained personnel would likely be lowering the false positivity rate of screening tests relative to a clinical endpoint of CIN2+, for example. Including clinical endpoints for at least a subset of the data would greatly strengthen this argument and the overall manuscript.

- Can the authors please clarify and/or elaborate on why 95% of the women eligible for this study were included in this study?

- Table 2 should include statistical analysis of differences between patient characteristics across categories, in addition to the values and percentages.

- It would be helpful if Tables 3 and 4 included the total number of people included in each of the predictor categories.

Reviewer #2: The authors present the prevalence and predictors of a positive VIA screen among women living with HIV in two different provinces of Eswatini, who access specific health facilities. It is a well-written article, and the analyses in it is sound. However, there are some issues to be addressed.

Introduction:

- There are missing citations in the introduction; multiple paragraphs written have only one or two citations when more are appropriate.

- There is a missing discussion of the literature on prevalence and predictors of positive VIA screens in similar contexts, and within Eswatini. (For ex., what has prior research found?)

- There is a missing discussion on predictors in other contexts.

- There is a missing discussion on the rationale and reasons for completing this analysis.

Methods:

- Why were the two provinces selected?

- How were the health centers selected?

- Line 208: “Casewise analysis was used… all patients had the outcome of interest.” This should be rewritten to indicate authors mean there were no missing observations for the outcome variable.

Results:

- I recommend re-arranging the way the results are presented for “Last VL Outcome” in a more ordered manner. (Undetectable – Suppressed – Unsuppressed – NO VL).

Discussion:

- The authors would benefit from adding a short, summary paragraph of their findings before delving into the discussion.

- There is an inadequate number of citations in the discussion, with some paragraphs having zero citations despite needing them.

6. PLOS authors have the option to publish the peer review history of their article (what does this mean?). If published, this will include your full peer review and any attached files.

**Do you want your identity to be public for this peer review?** For information about this choice, including consent withdrawal, please see our Privacy Policy.

Reviewer #1: No

Reviewer #2: No

---

## [Editor Report · Decision Letter 1]

6 Mar 2024

Cervical Cancer Screening Outcomes for HIV-positive Women in the Lubombo and Manzini regions of Eswatini – Prevalence and Predictors of a Positive Visual Inspection with Acetic Acid (VIA) Screen

PGPH-D-23-02447R1

Dear Dr Williams,

We are pleased to inform you that your manuscript 'Cervical Cancer Screening Outcomes for HIV-positive Women in the Lubombo and Manzini regions of Eswatini – Prevalence and Predictors of a Positive Visual Inspection with Acetic Acid (VIA) Screen' has been provisionally accepted for publication in PLOS Global Public Health.

Best regards,

Nebiyu Dereje, MPH, PhD

Academic Editor